# Referral pattern of oral and maxillofacial surgery cases in Sudan: A retrospective age- and sex-specific analysis of 3,478 patients over four years

**Musadak Ali Karrar Osman**[1]*, **Mohammed Hassan Ibrahem Aljezoli**[2], **Mohamed Alfatih Mohamed Alsadig**[3], **Ahmed Mohamed Suliman**[4]

1 Faculty of Dental Medicine and Surgery, Oral & Maxillofacial Department, National University, Khartoum, Sudan, 2 Oral and Maxillofacial, Saudi German Hospitals, Hail, Saudi Arabia, 3 Department of Orthodontics, College of Dentistry, University of The East, Manila, Philippines, 4 Faculty of Dentistry, Maxillofacial Surgery Department, University of Khartoum, Khartoum, Sudan

* musadak@hotmail.com

**Data Availability Statement:** All relevant data are within the paper and its Supporting Information files.

## Abstract

Oral and maxillofacial surgery (OMFS) is a specialty widening in its scope. An objective analysis of the referral pattern can provide essential information to improve healthcare. This four-year retrospective study was implemented in Khartoum Teaching Dental Hospital. Data (age, sex, diagnosis, and type of treatment) were collected from patient records. Disease frequency, as well as the effect of sex and age, were analyzed for each group. The frequency of treatment types was also assessed. Data were collected from a total of 3,478 patients over the four-year study period. There was a male predominance with the third decade of life being the most common age group. Pathological diseases were the most common (37%) reason for referral, followed by trauma (31%). Temporomandibular joint (TMJ) disorders and dentoalveolar extraction were the least frequently observed. Open reduction and internal fixation (ORIF) was the most commonly performed procedure (28%). These data represent the epidemiology of oral and maxillofacial diseases in Sudan. Given that the third decade of life is the most represented age group, it is beneficial to learn the long-term consequences of these diseases in these young patients and to use modern surgical techniques to improve their lives.

## Introduction

Oral and maxillofacial surgery (OMFS) is a specialty that is expanding in scope [1]. Despite the high number of services offered by OMFS specialists, few reports explore the overall characteristics of patients undergoing oral and maxillofacial procedures[2]. Most published articles subjectively assess the scope of OMFS practices [3] and focus on a particular type of service [2,4]. Overall, the lack of knowledge in this area limits health institutions' ability to provide better services and to improve their infrastructure. Thus, the importance of an epidemiological study to enhance the health care system is evident [5].

**Funding:** This research did not receive any specific grant from funding agencies in the public, commercial, or not-for-profit sectors.

**Competing interests:** The authors declare that they have no competing interests.

The department of OMFS in Khartoum Teaching Dental Hospital (KTDH) receives referrals from all parts of Sudan and treats a nondefined population. A lack of epidemiological research in oral and maxillofacial diseases in Sudan makes it a favorable location for such a study. Not only is the literature deficient, but little is known about the regional variations in the epidemiology of surgical diseases [6].

The present study aims to assess the prevalence of diseases diagnosed in a cohort of patients visiting an OMFS service in Sudan over four years, the pattern and distribution of OMFS diagnoses in relation to age and sex, and the type of treatments provided. The prevalence of specific disease groups can help to quantify the scope of OMFS surgery in Sudan, thus increasing awareness of the training needed to competently perform surgery.

## Material and methods

This is a retrospective study of the OMFS unit in KTDH. Patients attended the outpatient clinic between January 1, 2011, and December 31, 2014. Data were collected anonymously using patient records. All patients were categorized based on their age, sex, and diagnosis, as well as type of treatment. The diagnoses were coded according to the International Classification of Diseases (ICD-10). Then, the diagnosis was classified into seven major groups: pathology, trauma, infection, developmental defect, Temporomandibular joint (TMJ) disorders, abnormalities of teeth, and other. Other included diseases where nonsurgical treatment is the primary or only approach. Then, and whenever possible, Neville BW et al.'s textbook of Oral and Maxillofacial Pathology [7] nomenclature was adopted for individual diagnoses as well as for classification and subclassification, with minor modifications as necessary to avoid overlap (Table 1). Guided by the type of intervention, the treatment type was classified into 11 clinical categories.

Descriptive statistics were analyzed using SPSS version 24 (SPSS Inc., Chicago, IL, USA). The prevalence of disease groups, as well as the effect of age and sex, were evaluated. For comparison, the relevant existing literature was reviewed. Ethics approval was obtained from the KTDH research committee.

## Result

### Diseases

A total of 3,478 cases met the inclusion criteria. Of these, 63.1% (n = 2,196 cases) were male and 36.9% (n = 1,282) were female, with a ratio of 1.7:1. Age ranged from days to 95 years, with a mean age of 30.8 ± 20.2. The most common age group represented were those in the third decade of life 23.2% (n = 808), followed by the second decade 18.4% (n = 641) (Table 2). The mean age of males was 30.6 years and 31.2 for females, with no statistical significance difference ($p = 0.4$).

In accordance with ICD-10, there were 11 different clinical categories (Table 3).

The frequency of diagnosis groups is indicated in Fig 1. Except for the group "other" and "abnormality of teeth" males were more affected than females. Moreover, the discrepancy was obvious in the trauma group (Fig 2).

Regarding age, all diseases occurred at the third decade of life or earlier, except those grouped as "other," for which the sixth decade of life was most prevalent (Table 2).

### Pathologies

Diagnosed pathologies were the most common reason for patient visits, 36.1% (n = 1,256) (Fig 1), with the classification, including subclasses and diseases diagnosed are indicated in Fig 3 and Table 4, respectively.

**Table 1. The diagnosis groups, classification and subclassification of the study.**

**1. Pathology**
a. Benign
 Odontogenic
 Salivary gland
 Soft tissue
 Bone pathology
 Hematology
b. Malignant
 Odontogenic
 Salivary gland
 Epithelium
 Soft tissue
 Bone pathology
 Hematology
c. Cyst
 Odontogenic
 Salivary gland
 Nonodontogenic
d. Inflammatory

**2. Trauma**

**3. Infection**
. Odontogenic infection
. Nonodontogenic infection

**4. Developmental defect**
. Cleft deformity
. Skeletal deformity
. Inherited and syndromic disorder
. Soft tissue deformity

**5. Temporomandibular joint disorders**

**6. Abnormalities of teeth**

**7. Other**
. Inflammatory
. Autoimmune disorder
. Facial pain and neuromuscular diseases
. Salivary gland
. Epithelium
. Bone pathology

**Table 2. Distribution of oral and maxillofacial diseases by age (years).**

| Disease groups | Age | | | | | | | | Total |
|---|---|---|---|---|---|---|---|---|---|
| | **0–9** | **10–19** | **20–29** | **30–39** | **40–49** | **50–59** | **60–69** | **≥ 70** | |
| **Trauma** | 61 | 181 | 403 | 238 | 111 | 60 | 35 | 19 | 1108 |
| **Developmental defect** | 287 | 124 | 47 | 20 | 10 | 2 | 2 | 0 | 492 |
| **Pathology** | 107 | 233 | 203 | 150 | 150 | 144 | 122 | 147 | 1256 |
| **Temporomandibular joint disorders** | 14 | 40 | 24 | 7 | 5 | 4 | 1 | 1 | 96 |
| **Infection** | 17 | 37 | 75 | 49 | 35 | 22 | 19 | 20 | 274 |
| **Other** | 7 | 21 | 29 | 27 | 28 | 30 | 31 | 22 | 195 |
| **Abnormalities of teeth** | 0 | 5 | 27 | 15 | 6 | 3 | 1 | 0 | 57 |
| **Total** | 493 (14.2%) | 641 (18.4%) | 808 (23.2%) | 506 (14.5%) | 345 (9.9%) | 265 (7.6%) | 211 (6.1%) | 209 (6.0%) | 3478 |

Data are shown as frequency (percentage).

P-value = 0.001.

**Table 3. Diagnosis distribution by ICD-10 classification.**

| ICD-10 classification | Frequency | Percentage |
|---|---|---|
| 1. Injury, poisoning and certain other consequences of external causes | 1114 | 32.0% |
| 2. Diseases of the digestive system | 760 | 21.9% |
| 3. Congenital malformations, deformations and chromosomal abnormalities | 472 | 13.6% |
| 4. Benign neoplasms | 438 | 12.6% |
| 5. Malignant neoplasms | 405 | 11.6% |
| 6. Diseases of the musculoskeletal system and connective tissue | 116 | 3.3% |
| 7. Diseases of the nervous system | 65 | 1.9% |
| 8. Diseases of the skin and subcutaneous tissue | 51 | 1.5% |
| 9. Certain infectious and parasitic diseases | 28 | 0.8% |
| 10. Diseases of the respiratory system | 22 | 0.6% |
| 11. External causes of morbidity and mortality | 7 | 0.2% |
| Total | 3478 | 100.0% |

Both sex and age groups had a significant relationship in terms of classification distribution (Table 5). Males were more affected than females, and the second decade of life was the most affected age group. Regarding sex, males had more malignant diseases, while females had more benign diseases. Regarding age groups, the most common pathology was cystic diseases in the first or second decade of life. In contrast, the third and fourth decades of life featured a higher frequency of benign neoplasms. From the fifth decade of life and onwards, the most common diseases were malignancies (Table 5).

**Benign.** The most common class was benign diseases 35.1% (n = 441). The odontogenic tumor was the most common frequent subgroup 33.3% (n = 147), followed by bone pathology

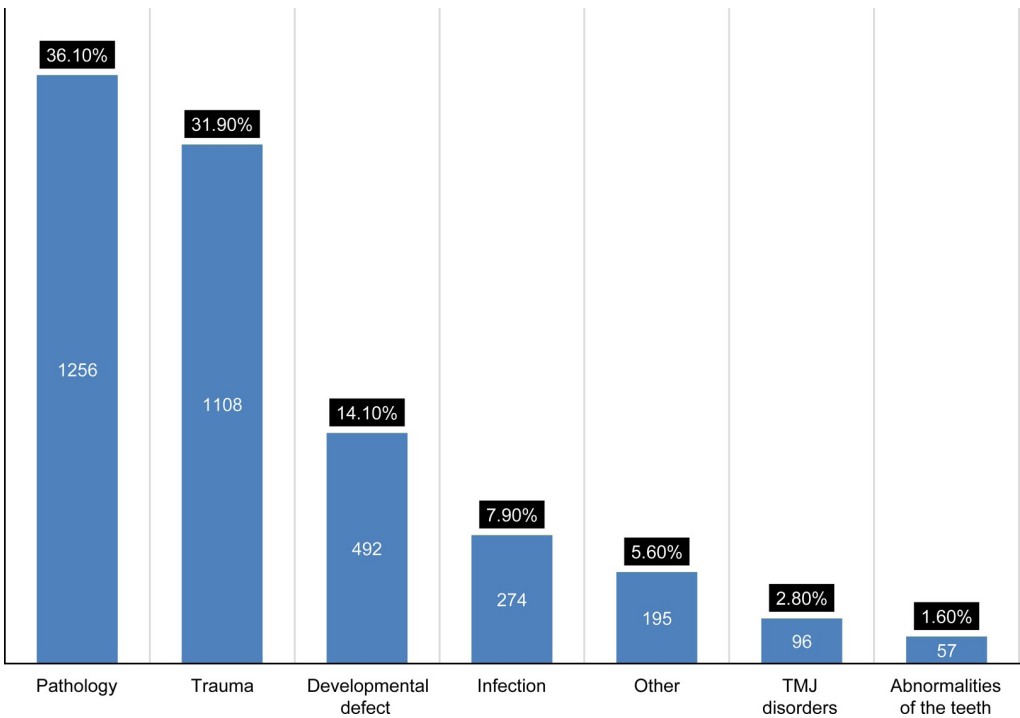

**Fig 1. Distribution (number and percentage) of the diagnosis groups.**

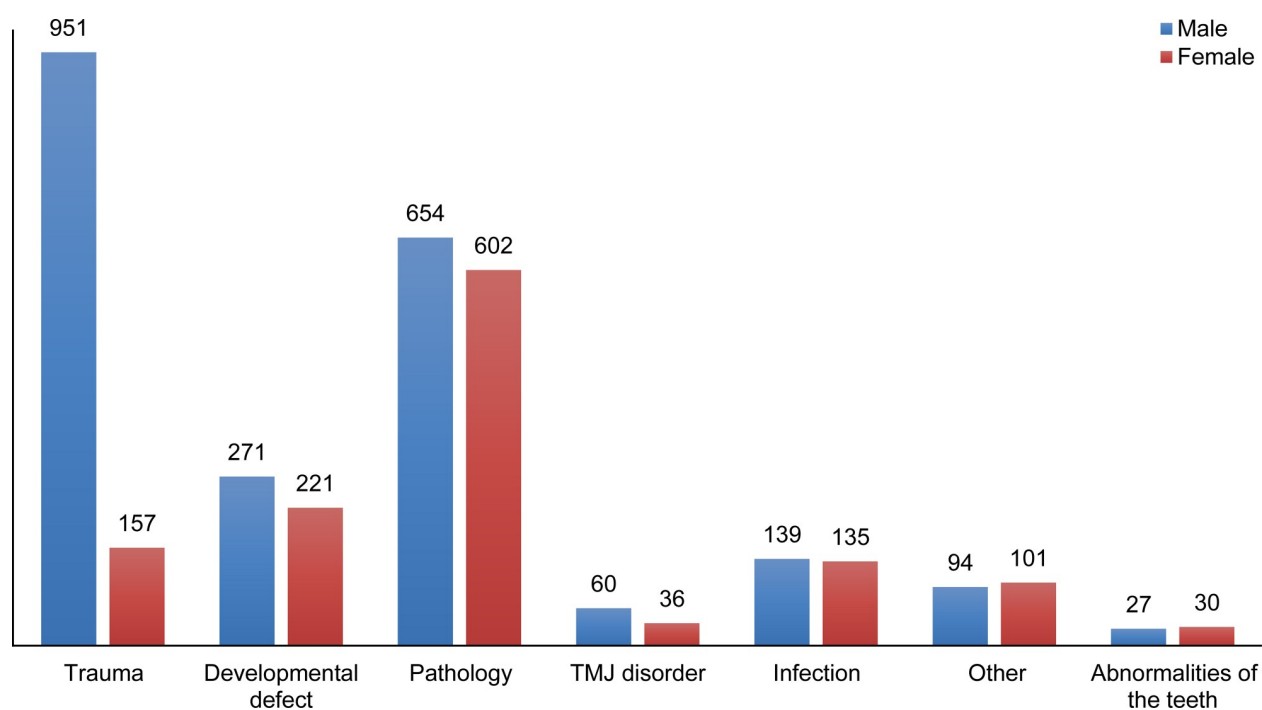

**Fig 2. Distribution (number) of disease groups by sex.**

in the form of nonodontogenic tumor 24.9% (n = 110 cases). Ameloblastoma was the most common individual diagnosis 75% (n = 110) (Table 4). Pleomorphic adenoma was the second most common individual diagnosis with 76 cases, representing 95% of benign salivary gland

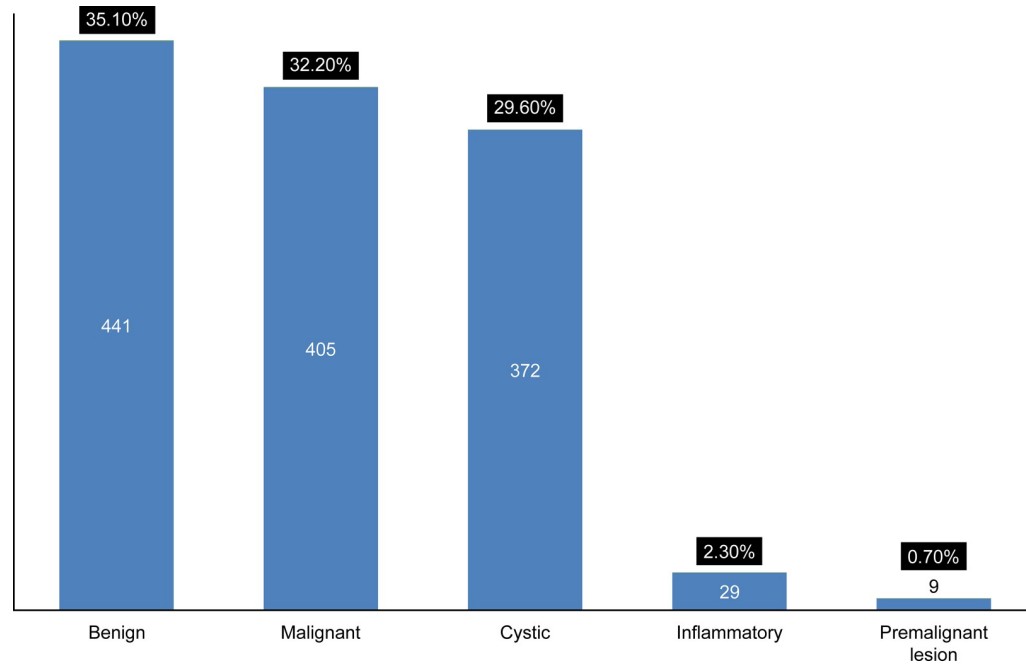

**Fig 3. Distribution (number and percentage) of pathological disease classification.**

**Table 4. Frequency of 1256 pathological diseases presented as class, subclass, and individual diagnosis.**

| Benign: 441 (35.1%) | Frequency | Percentage within subclass |
|---|---|---|
| *Odontogenic: 146 cases* | | |
| Ameloblastoma | 110 | 75% |
| Odontoma | 19 | 13% |
| Odontogenic myxoma | 10 | 7% |
| Calcifying epithelial odontogenic tumor | 4 | 3% |
| Ameloblastic fibroma | 1 | 1% |
| Squamous odontogenic tumor | 1 | 1% |
| Ameloblastic fibro-odontoma | 1 | 1% |
| *Salivary gland: 80 cases* | | |
| Pleomorphic adenoma | 76 | 95% |
| monomorphic adenoma | 3 | 4% |
| Myoepithelioma | 1 | 1% |
| *Soft tissue: 104 cases* | | |
| Hemangioma | 25 | 24% |
| Pyogenic granuloma | 19 | 18% |
| Lipoma | 16 | 15% |
| Vascular malformation | 15 | 14% |
| Lymphangioma | 10 | 10% |
| Fibroma | 6 | 6% |
| Neurofibroma | 5 | 5% |
| Fibroepithelial polyp | 2 | 2% |
| Peripheral giant cell granuloma | 1 | 1% |
| Schwannoma | 1 | 1% |
| Angiofibroma | 1 | 1% |
| Fibromatosis | 1 | 1% |
| Solitary fibrous tumor | 2 | 2% |
| *Bone pathology: 110 cases* | | |
| Ossifying fibroma | 50 | 45% |
| Florid cemento-osseous dysplasia | 20 | 18% |
| Fibrous dysplasia | 20 | 18% |
| Central giant cell granuloma | 13 | 12% |
| Chondroma | 5 | 5% |
| Brown tumor | 2 | 2% |
| *Hematology: 1 case* | | |
| Eosinophilic granuloma | 1 | 100% |
| **Malignant: 405(32.2%)** | | |
| *Odontogenic: 4 cases* | | |
| Malignant ameloblastoma | 1 | 25% |
| Ghost cell odontogenic carcinoma | 1 | 25% |
| Ameloblastic carcinoma | 1 | 25% |
| Malignant odontogenic myxoma | 1 | 25% |
| *Salivary gland:61 cases* | | |
| Mucoepidermoid carcinoma | 23 | 38% |
| Adenoid cystic carcinoma | 16 | 26% |
| Adenocarcinoma, not otherwise specified (NOS) | 15 | 25% |
| Acinic cell adenocarcinoma | 5 | 8% |
| Basal cell adenocarcinoma | 1 | 2% |

*(Continued)*

**Table 4.** (Continued)

| Benign: 441 (35.1%) | Frequency | Percentage within subclass |
|---|---|---|
| *Odontogenic: 146 cases* | | |
| Myoepithelial carcinoma | 1 | 2% |
| *Epithelium: 304 cases* | | |
| Squamous cell carcinoma | 274 | 90% |
| Verrucous carcinoma | 13 | 4% |
| Clear cell carcinoma | 5 | 2% |
| Malignant melanoma | 5 | 2% |
| Basal cell carcinoma | 4 | 1% |
| Nasopharyngeal carcinoma | 3 | 1% |
| *Soft tissue: 16 cases* | | |
| Fibrosarcoma | 7 | 44% |
| Metastatic lesion | 3 | 19% |
| Rhabdomyosarcoma | 2 | 13% |
| Malignant fibrous histiocytoma | 1 | 6% |
| Hemangioendothelioma | 1 | 6% |
| Liposarcoma | 1 | 6% |
| Angiosarcoma | 1 | 6% |
| *Bone pathology: 8 cases* | | |
| Ewing's sarcoma | 4 | 50% |
| Osteosarcoma | 2 | 25% |
| Chondrosarcoma | 2 | 25% |
| **Hematology: 12 cases** | | |
| Lymphoma | 11 | 92% |
| Leukemia | 1 | 8% |
| **Cyst: 372(29.6%)** | | |
| *Odontogenic: 171 cases* | | |
| Dentigerous cyst | 66 | 39% |
| Radicular cyst | 54 | 32% |
| Odontogenic keratocyst | 44 | 26% |
| Lateral periodontal cyst | 4 | 2% |
| Calcifying odontogenic cyst | 3 | 2% |
| *Salivary gland:129 cases* | | |
| Ranula | 116 | 90% |
| Mucocele | 13 | 10% |
| *Nonodontogenic: 72 cases* | | |
| Nasopalatine cyst | 43 | 60% |
| Dermoid/epidermoid cyst | 11 | 15% |
| Globulomaxillary cyst | 7 | 10% |
| Thyroglossal cyst | 5 | 7% |
| Median palatine cyst | 3 | 4% |
| Aneurysmal bone cyst | 2 | 3% |
| Lymphoepithelial cyst | 1 | 1% |
| **Inflammatory: 29(2.3%)** | | |
| Salivary gland stone | 28 | 97% |
| Necrotizing sialometaplasia | 1 | 3% |
| **Premalignant diseases: 9 (0.7%)** | | |
| Leukoplakia/ Erythroplakia | 9 | 100% |

**Table 5. Pathological disease groups distribution by age (years) and sex.**

| Pathological diseases groups | Age | | | | | | | | Sex | | Total |
|---|---|---|---|---|---|---|---|---|---|---|---|
| | 0–9 | 10–19 | 20–29 | 30–39 | 40–49 | 50–59 | 60–69 | ≥ 70 | Male | Female | |
| Benign | 33 | 92 | 101 | 75 | 53 | 42 | 19 | 26 | 197 | 244 | 441 |
| Malignant | 10 | 10 | 24 | 25 | 53 | 85 | 87 | 111 | 244 | 161 | 405 |
| Inflammatory | 0 | 4 | 0 | 5 | 11 | 6 | 1 | 2 | 15 | 14 | 29 |
| Cystic | 64 | 127 | 77 | 45 | 31 | 10 | 11 | 7 | 191 | 181 | 372 |
| Premalignant diseases | 0 | 0 | 1 | 0 | 2 | 1 | 4 | 1 | 7 | 2 | 9 |
| Total | 107 (8.5%) | 233 (18.6%) | 203 (16.2%) | 150 (11.9%) | 150 (11.9%) | 144 (11.5%) | 122 (9.7%) | 147 (11.7%) | 654 (52.1%) | 602 (47.9%) | 1256 |

Data are shown as frequency (percentage).

P-value = 0.001.

neoplasms (Table 4). The third decade of life was the most involved age group 22.9% (n = 101), and females were the predominant sex 55.3% (n = 244) (Table 5).

**Malignant.** Malignant diseases were the second most common pathological diagnosis 32.2% (n = 405). Squamous cell carcinoma was the most common individual diagnosis with 272 cases, representing 90.1% of malignant epithelial neoplasms. Mucoepidermoid carcinoma was the second most common individual malignant diseases 38% (n = 23) regarding malignant salivary gland neoplasms (Table 4). This type of neoplasm rarely occurs before the fifth decade of life, but it drastically increases in incidence during the subsequent decade, and males are the predominant sex 60.2% (n = 244) (Table 5).

**Cystic.** A total of 372 cases (29.6%) were diagnosed with cysts. Of these, 45.9% (n = 171) were odontogenic cysts, 34.6% 9 (n = 129) were sialoceles, and 19.3% (n = 72) were nonodontogenic cysts. Among odontogenic cysts, dentigerous cysts were the most common 39% (n = 66), while nasopalatine cysts were the most common nonodontogenic cyst 60% (n = 43) (Table 4). This type of pathology occurs more frequently during the first three decades of life and rarely occurs in older age groups. Males show slightly higher frequency 51.3% (n = 191) 51.3% (Table 5)

The frequency of premalignant diseases, as well as their distribution among age groups and sex, are indicated in Tables 4 and 5, respectively.

## Trauma

Trauma was the second most common class 31.8% (n = 1108) (Fig 1). Males were significantly more involved 85.8% (n = 951) with the third decade of life being the most represented age group 36.4% (n = 403) (Table 2).

## Developmental defects

Developmental defects represented 14.1% (n = 492) of the study sample (Fig 1). The most common defect was clefts 86.6% (n = 426), with a predominance of cleft lips noted 46.4% (n = 197), followed by cleft lips and palates 28.2% (n = 120) and cleft palates 25.4% (n = 108), (Table 6).

Among all cleft cases, only 61% (n = 260) were treated during the first decade of life (Table 7). Cleft palates were more frequent in females 52.8% (n = 57), while cleft lips and palates and cleft lips were more predominant in males 54.2% (n = 65), and 60.1% (n = 19) respectively (Table 7).

**Table 6. Frequency of 492 developmental defects.**

| Cleft deformity: 426 cases | Frequency | Percentage within subclass |
|---|---|---|
| Cleft lip | 198 | 46% |
| Cleft lip and palate | 120 | 28% |
| Cleft palate | 108 | 25% |
| *Skeletal deformity: 24 cases* | | |
| Class II malocclusion | 9 | 38% |
| Condylar hyperplasia | 7 | 29% |
| Class III mal occlusion | 3 | 13% |
| Hemifacial hyperplasia | 2 | 8% |
| Severe skeletal anterior open bite | 2 | 8% |
| Mandible retrognathia with bifid tongue and cleft mandible | 1 | 4% |
| *Inherited and syndromic disorder: 10 cases* | | |
| Osteopetrosis | 3 | 30% |
| Cherubism | 2 | 20% |
| Gorlin syndrome | 1 | 10% |
| Melkersson–Rosenthal syndrome | 1 | 10% |
| Osteogenesis imperfecta | 1 | 10% |
| Ectodermal dysplasia | 1 | 10% |
| Xeroderma pigmentosa | 1 | 10% |
| *Soft tissue deformity: 32 cases* | | |
| Ankyloglossia | 28 | 88% |
| Double lip | 3 | 9% |
| Geographic tongue | 1 | 3% |

## Infections

Over the study period, a total of 7.9% (n = 274) of patients were affected by orofacial infections. Of these, 78.8% (n = 216) and 21.1% (n = 58) were odontogenic and nonodontogenic in origin, respectively (Table 8). The most affected age group were those in their third decade of life with 28.2% (n = 61) and 24.1% (n = 14) being odontogenic and nonodontogenic, respectively (*p*-value = 0.03).

In our sample, females represented 50.5% (n = 109) of odontogenic infections and males represented 55.1% (n = 32) of nonodontogenic infections; however, the sex difference was not statistically significant (p-value = 0.27).

**Table 7. Frequency and cleft subclassification Distribution by age group and sex.**

| Cleft subclassification | Age | | | | | | | Sex | | Total |
|---|---|---|---|---|---|---|---|---|---|---|
| | 0–9 | 10–19 | 20–29 | 30–39 | 40–49 | 50–59 | 60–69 | Male | Female | |
| **Cleft lip** | 113 | 57 | 15 | 8 | 4 | 1 | 0 | 119 | 79 | 198 |
| **Cleft Lip and palate** | 85 | 23 | 7 | 2 | 3 | 0 | 0 | 65 | 55 | 120 |
| **Cleft Palate** | 62 | 28 | 13 | 2 | 1 | 1 | 1 | 51 | 57 | 108 |
| **Total** | 260 (61.0%) | 108 (25.4%) | 35 (8.2%) | 12 (2.8%) | 8 (1.9%) | 2 (0.5%) | 1 (0.2%) | 235 (55.2%) | 191 (44.8%) | 426 |

Data are shown as frequency (percentage).

P-value (age) = 0.23.

P-value (sex) = 0.09.

**Table 8. Frequency of 274 infections.**

| *Odontogenic infection*: *216* | Frequency | Percentage within subclass |
|---|---|---|
| Single space | 53 | 25% |
| Osteomyelitis | 50 | 23% |
| multiple/deep space | 49 | 23% |
| Ludwig's angina | 45 | 21% |
| Necrotizing fasciitis | 19 | 9% |
| *Nonodontogenic infection*: *58 cases* | | |
| Infected plate | 18 | 31% |
| Tuberculous cervical lymphadenitis | 6 | 10% |
| Mumps | 5 | 9% |
| Leishmaniasis | 4 | 7% |
| Aspergillosis | 4 | 7% |
| Cancrum oris | 3 | 5% |
| Osteoradionecrosis | 3 | 5% |
| Tuberculous osteomyelitis | 2 | 3% |
| Candidiasis | 2 | 3% |
| Actinomycosis | 2 | 3% |
| Carbuncle | 1 | 2% |
| Viral warts | 1 | 2% |
| HIV | 1 | 2% |
| Histoplasmosis | 1 | 2% |
| Herpetic gingivostomatitis | 1 | 2% |
| Oral warts | 1 | 2% |
| Syphilitic ulcer | 1 | 2% |
| Squamous papilloma | 1 | 2% |
| Shingle | 1 | 2% |

## TMJ disorders

Fig 4 illustrates the frequency of individual TMJ disorders, corresponding to a total of 2.8% (n = 96) of the study sample. The most frequent diagnosis was ankylosis 82.3% (n = 79).

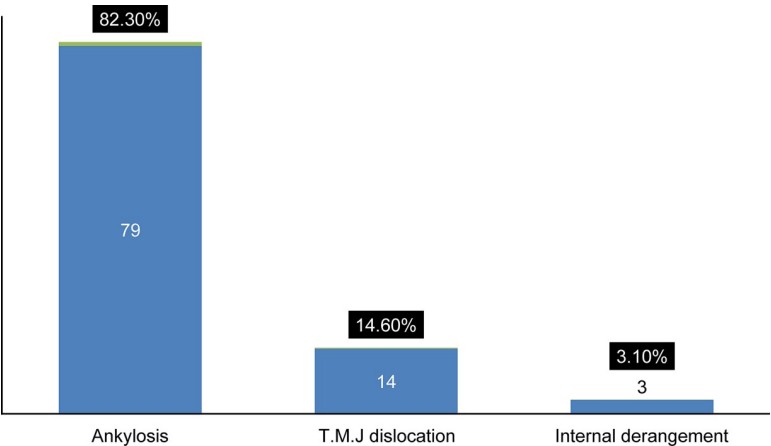

**Fig 4. Distribution (number and percentage) of temporomandibular joint disorders.**

The second decade of life was the most represented age group for ankylosis as well as dislocation deformities with 45.6% (n = 36) and 28.5% (n = 4), respectively (*p*-value = 0.001). There was a female predominance in dislocation 57.1% (n = 8) and internal derangement 100% (n = 3), whereas ankylosis demonstrated a male predominance 68.3% (n = 54); *p*-value = 0.015.

## Abnormalities of teeth

A 1.6% (n = 57) accounted for impacted teeth which indicated for extraction under general anesthesia (Fig 1). The effects of sex and age are indicated in Fig 2 and Table 2.

**Other.** Table 9 shows the distribution of the "other oral diseases" class. The effects of sex and age are indicated in Table 10.

## Type of treatment

The most commonly performed surgery was open reduction and internal fixation (ORIF; 779 cases: 22.4%), followed by tumor excision ± neck dissection (694 cases: 20%). In fact, 83 neck

**Table 9. Frequency of 195 other class diseases.**

| *Facial Pain and Neuromuscular Diseases*: 92 cases | Frequency | Percentage |
|---|---|---|
| Trigeminal neuralgia | 35 | 38% |
| Myofascial  pain | 23 | 25% |
| Facial palsy | 18 | 20% |
| Atypical facial pain | 10 | 11% |
| Massetric hypertrophy | 3 | 3% |
| Burning mouth syndrome | 3 | 3% |
| *Autoimmune disorder*: 59 cases | | |
| Erythema multiform | 17 | 29% |
| Vesiculobullous lesions | 16 | 27% |
| Sjögren syndrome | 9 | 15% |
| Aphthous ulcer | 8 | 14% |
| Lichen planus | 5 | 8% |
| Stevens–Johnson syndrome | 2 | 3% |
| Angioedema | 1 | 2% |
| Erythema bullosa haemorrhagica | 1 | 2% |
| *Salivary gland*: 30 cases | | |
| Sialadenitis | 22 | 73% |
| Mucositis | 1 | 3% |
| Cheilitis glandularis | 1 | 3% |
| Xerostomia | 3 | 10% |
| Sialorrhea | 1 | 3% |
| Frey's syndrome | 1 | 3% |
| Sialosis | 1 | 3% |
| *Epithelium*: 13 cases | | |
| Heck's disease | 6 | 46% |
| Eosinophilic traumatic ulcer | 4 | 31% |
| Squamous hyperplasia | 1 | 8% |
| Naevous | 1 | 8% |
| Frictional keratosis | 1 | 8% |
| *Bone pathology: 1 case* | | |
| Paget's disease | 1 | 100% |

Table 10. Sex and age effect on other class.

| Other class subclassification | Age | | | | | | | | Sex | | Total |
|---|---|---|---|---|---|---|---|---|---|---|---|
| | 0–9 | 10–19 | 20–29 | 30–39 | 40–49 | 50–59 | 60–69 | ≥70 | Male | Female | |
| Inflammatory | 1 | 5 | 4 | 4 | 3 | 4 | 0 | 3 | 13 | 11 | 24 |
| Autoimmune disorder | 1 | 10 | 6 | 7 | 13 | 11 | 9 | 2 | 25 | 34 | 59 |
| Facial pain and neuromuscular diseases | 3 | 3 | 16 | 15 | 11 | 14 | 17 | 13 | 43 | 49 | 92 |
| Salivary gland | 0 | 1 | 1 | 1 | 1 | 0 | 1 | 1 | 3 | 3 | 6 |
| Epithelium | 2 | 2 | 2 | 0 | 0 | 1 | 3 | 3 | 10 | 3 | 13 |
| Bone pathology | 0 | 0 | 0 | 0 | 0 | 0 | 1 | 0 | 0 | 1 | 1 |
| Total | 7 (3.6%) | 21 (10.8%) | 29 (14.9%) | 27 (13.8%) | 28 (14.4%) | 30 (15.4%) | 31 (15.9%) | 22 (11.3%) | 94 (48.2%) | 101 (51.8%) | 195 |

p-value (age) = 0.205.
p-value (sex) = 0.264.

dissections were recorded either alone or as part of another operation. Table 11 shows the frequency of each type of treatment. The category "others" includes surgeries performed in less than 1% of patient visits, namely the surgical removal of teeth, soft tissue repair, palate removal, and rarely orthognathic surgery, coronoidectomy, bone shaving, and relocation of the submandibular ducts. Only two patients refused treatment.

## Discussion

Hospital-based studies are the closest to clinical practice and represent its standards [8]. The current study evaluates the scope of OMFS in Sudan. Pathological diseases accounted for most of the diagnoses (36.1%), followed by traumatic events (31.9%). The frequency of the diseases differs between studies; however, it fluctuates between trauma and pathological diseases [8,9], which depends on the scope of focus for each hospital as well as the study population. While hospitals with an elective procedural scope tend to have more developmental deformities and pathological diseases, trauma is more common in hospitals possessing an emergency scope [2,8,10]. Furthermore, studies focused on outpatient populations may differ compared to inpatient focused studies [11]. The OMFS department in KTDH is the biggest in Sudan and is a

Table 11. Type of treatment.

| Procedure | Frequency | Percentage |
|---|---|---|
| 1. Open Reduction and internal Fixation (ORIF) | 779 | 22.4 |
| 2. Tumor excision ± neck dissection | 694 | 20.0 |
| 3. Cleft lip/palate repair | 426 | 12.2 |
| 4. Conservative management/follow up | 302 | 8.7 |
| 5. Enucleation | 254 | 7.3 |
| 6. Incision & drainage/debridement/sequestrectomy | 250 | 7.2 |
| 7. Other | 164 | 4.7 |
| 8. Salivary gland excision | 214 | 6.2 |
| 9. Close reduction | 191 | 5.5 |
| 10. Temporomandibular arthroplasty | 96 | 2.8 |
| 11. Biopsy | 108 | 3.1 |
| Total | 3478 | 100 |

highly equipped maxillofacial center. Furthermore, it is the only department in Sudan that receives referrals for a nondefined population, distinguishing it from military and police hospitals that focus on their personnel or employees.

Most patients were male, which aligns with previous reports [8,9,12]). Higher prevalence of tobacco use with subsequent malignancy and involvement in trauma may exhibit a sex norm pattern for males. Sex norms are defined as socially constructed roles, behaviors, activities, and attributes that a given society considers appropriate for men and women [13]. In contrast, Bezerra et al. [2] and Brennan et al. [4] report that patients are predominantly female. Age ranges from days to 95 years, with the mean age of 30.84 ± 20.17 years. The second and third decades of life were the most common age groups represented.

The frequency of different diseases is somewhat determined by the age distribution of patients seen.

While developmental defects and odontogenic tumors and cysts are more prevalent in younger age groups, squamous cell carcinoma is more prevalent in the elderly. Moreover, Ibikunle et al. [9] noted the skew distribution of age, and explained by inclusion of minor surgeries, for instances routine dental extraction.

## Pathologies

**Benign.** The most common pathological diagnoses were benign diseases, representing 33.9% of all diagnoses. Ameloblastoma was the most frequent diagnosis. This finding is in agreement with reports by Adebayo et al. [9] and Siriwardena et al [14], where benign diseases were assessed along with benign and odontogenic tumors, respectively, and they were found to be the most common among both categories. Pleomorphic adenoma was the second most common benign neoplasm and the most common among benign salivary gland neoplasms. This finding is consistent with Vargas et al.'s [15].

**Malignant.** In terms of malignant diseases, oral squamous cell carcinoma (OSCC) is the most common diagnosis, with 272 cases (67%). In fact, if individual diagnoses are considered, OSCC represents the most common presentation after trauma. This high frequency rate can be explained by the results of a recent metanalysis findings, where OSCC significantly correlated to "toombak," a local name of Sudanese smokeless tobacco [16]. Furthermore, in vitro, research indicates that toomback damages human oral epithelium DNA and leads to malignant transformation [17]. Knowing that, and the fact that approximately 45% of men aged 40 years or older use toombak [18], there is an urgent need for programs addressing early detection and prevention. Interestingly, there is no statistical relationship between oral cancer and snus, the Swedish smokeless tobacco. This disparity is correlated to higher concentration (100-fold) of N-nitrosamines, the most abundant carcinogens, in Sudanese toombak [18]. Moreover, snus has been suggested to play a harm-reduction role by providing an alternative to cigarettes and reducing the smoking incidence as well as the related mortality and morbidity [19,20]. Such a strategy may be considered as one preventive method.

**Cystic.** In this study, dentigerous cysts were the most observed odontogenic cysts, which is consistent with Butt et al [21]. In contrast, the most frequent odontogenic cyst reported in the literature is radicular cyst [22–24], which can be attributed to the fact that most patients treated for periapical cysts are treated as outpatients in minor-surgery departments and are rarely sent for histopathological examination. Furthermore, many opt for a nonsurgical endodontics approach and leave surgical intervention as the last option [25], and we may experience a low volume of such cases in the future. Although sialoceles are the most common individual diagnosis among cystic diseases in the present study, the authors have no justification for that, and it should be considered in further research.

## Trauma

Maxillofacial injuries usually constitute a high workload for oral and maxillofacial surgeons [8]. In the present study, maxillofacial injury comprised 31.8% of all patients. A clear predominance (86%) of males is noted, and the third decade of life is the most represented age group. The same trend is indicated in other studies [26,27]. However, the sex difference is narrower in developed countries when compared to developing countries [28]. The fact that males are more likely to be involved in traffic accidents and interpersonal violence, this helps explain the higher frequency of trauma among males. Also, reckless driving behaviors and more physically active hobbies make young adults more likely to be involved in trauma [29,30].

## Developmental defects

Developmental defects represent 13.9% of all cases, with cleft deformity being the most common (87.6%). This prevalence confirms previous reports that the most common orofacial deformity is a cleft deformity [31]. In contrast, when including asymptomatic defects, fissured tongue, the cleft deformity frequency may decrease to 45% [11]. Generally, cleft lip with or without cleft palate (CL ± P) is believed to be etiologically related, while cleft palate (CP) is considered as a different category [7]. The importance of clinical presentations of the cleft lies in their inference regarding associated or potential defects. Many subjects with cleft lip and palate (CLP) had isolated defects, while most subjects with CP only had additional associated congenital anomalies [32]. Furthermore, CLP has the potential to develop skeletal malocclusion with a severity associated with the number of previous surgeries [33].

The pattern of cleft lip (CL; 46.4%), CLP (28.2%) and CP (25.4%) identified in the present study differs when compared to reports from Ethiopia, Hawaii, and Nigeria, where CLP occurred more frequently [32,34,35]. All reports indicate that CP occurs least often. However, the cleft deformity frequency reports from Sudan show inconsistent data. Ali et al. [36] reported the same trend as in the present study, and was a hospital-based study estimating that CLP was most frequent, and CL was the least frequent cleft deformity [37]. It is important to note that the lack of a national registry program for clefts makes pattern evaluation difficult in Sudan.

Notably, 39% of cleft patients were older than 10 years. The same trend was reported from other developing countries [38]. A possible cause is low socioeconomic status and poor access to health facilities. However, further investigation is needed to determine the relevance of factors such as delays in treatment-seeking behaviors.

In general, a male prevalence was noted in cleft deformity (55.2%). The same pattern was observed in Nigeria, Ethiopia, and Poland [39–41]. The frequency of CL ± P in males is higher when compared to females, whereas the frequency of CP alone is higher in females, consistent with the findings of Martelli et al. [42].

Another interesting observation is the low frequency of maxillofacial syndromic patients, which can be attributed to a lack of awareness of the role of the maxillofacial surgery specialty among the population and the medical community in the treatment of such lesions.

## Infections

Odontogenic infections are important because of their high incidence and associated morbidity [43] when compared to nonodontogenic infections. This kind of infection can spread to facial and deep neck spaces, can lead to cavernous sinus thrombosis, necrotizing fasciitis, descending mediastinitis, and can progress to sepsis [44,45]. Moreover, the incidence of deep neck space infection is reported to be significantly higher in patients with odontogenic infections when compared to nonodontogenic infections [46]. In the present study, the majority of

infections originated from odontogenic sources, which is consistent with findings reported by Igoumenakis et al. [47]. Of these infections, only 24.5% were single space infections, whereas the rest were complicated cases—namely, cases of multiple spaces, Ludwig's angina, and cervicofacial necrotizing fasciitis. Knowing that the severity of infections is correlated to the number and the site of the infected spaces [48], it is crucial to further study this type of patient and the reasons for late presentation.

The distribution of orofacial infection was higher in the third decade of life, with 55 cases (20.8%) and near equal disruption between the sexes. The same trend was noted in other developing countries [49].

## TMJ disorders

Diseases requiring TMJ surgeries are relatively rare, and the current study is no exception, with only 2.8% diagnosed. Kamalkumar et al. reported 1.2% TMJ surgical cases, and Islam et al. reported 15.1% TMJ ankylosis cases [8]. Most TMJ disorders in the study were TMJ ankylosis (82.3%), and TMJ dislocation represents only 14.6% of cases. However, as acute dislocation is usually treated in the outpatient department, the current estimate reflects only those cases that required further treatment after a failed outpatient intervention.

The results of this study are consistent with another study from Sudan [50], which reported that ankylosis more commonly occurred in adolescents. The associated deformities are more complicated. The technical difficulties in surgery and the high relapse rate [51] mandate a surgical protocol for future audit to arrive at best practices for Sudanese patients.

## Abnormalities of teeth

The low frequency of extractions under general anesthesia demonstrated in the present study does not reflect their actual number, as most extractions are performed in minor surgical departments and are not included in the referral clinics record.

## Type of treatment

The most common treatment modality was ORIF (768 cases: 22.1%), followed by tumor excision ± neck dissection (694 cases: 19.9%), including both benign and malignant diseases. Given that most patients with malignancy showed up late [52], neck dissection was performed in at least 83 cases. Although fracture may be treated quickly, bone and soft tissue resections may need full oral rehabilitation to improve quality of life. Likewise, CLP and TMJ ankylosis in young patients may necessitate additional surgeries in the future, such as orthognathic surgery and TMJ prosthesis.

## Conclusion

This is the first study presenting the overall prevalence and scope of OMFS in Sudan. Given that malignant diseases and trauma are frequently encountered, both can be addressed better by implementing preventive methods. Delayed care-seeking behaviors need more attention to identify causative factors and to support patients and their families. Furthermore, it provides a guide to improve provider education and to appropriately allocate health resources. However, it carries the limitation of a retrospective study in terms of inaccurate or incomplete data. Given the difficulties encountered during data collection, the using of a predesigned chart templates with each disease category may help in future research and auditing to improve the quality of OMFS Department services.

## Supporting information

**S1 File. Dataset.**
(XLS)

## Acknowledgments

The authors thank the KTDH's Medical Records Department for allowing access to records and data collection.

## Author Contributions

**Conceptualization:** Musadak Ali Karrar Osman, Mohamed Alfatih Mohamed Alsadig, Ahmed Mohamed Suliman.

**Data curation:** Musadak Ali Karrar Osman, Mohamed Alfatih Mohamed Alsadig.

**Formal analysis:** Musadak Ali Karrar Osman, Mohamed Alfatih Mohamed Alsadig.

**Methodology:** Mohammed Hassan Ibrahem Aljezoli.

**Validation:** Musadak Ali Karrar Osman.

**Writing – original draft:** Musadak Ali Karrar Osman, Mohamed Alfatih Mohamed Alsadig.

**Writing – review & editing:** Musadak Ali Karrar Osman, Mohammed Hassan Ibrahem Aljezoli, Mohamed Alfatih Mohamed Alsadig, Ahmed Mohamed Suliman.

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
