## [Decision Letter · Decision Letter 0]

22 Jan 2021

PONE-D-20-35051

Referral pattern of oral and maxillofacial surgery cases in Sudan: A retrospective age- and sex-specific analysis of 3,478 patients over four years

PLOS ONE

Dear Dr.  Osman

Thank you for submitting your manuscript to PLOS ONE. After careful consideration, we feel that it has merit but does not fully meet PLOS ONE’s publication criteria as it currently stands. Therefore, we invite you to submit a revised version of the manuscript that addresses the points raised during the review process.

We look forward to receiving your revised manuscript.

Kind regards,

Amit Sapra

Academic Editor

PLOS ONE

Journal Requirements:

2.We note that you have indicated that data from this study are available upon request. PLOS only allows data to be available upon request if there are legal or ethical restrictions on sharing data publicly. For information on unacceptable data access restrictions, please see http://journals.plos.org/plosone/s/data-availability#loc-unacceptable-data-access-restrictions.

Reviewers' comments:

Reviewer's Responses to Questions

**Comments to the Author**

1. Is the manuscript technically sound, and do the data support the conclusions?

Reviewer #1: Yes

2. Has the statistical analysis been performed appropriately and rigorously? 

Reviewer #1: Yes

3. Have the authors made all data underlying the findings in their manuscript fully available?

Reviewer #1: Yes

4. Is the manuscript presented in an intelligible fashion and written in standard English?

Reviewer #1: Yes

5. Review Comments to the Author

Reviewer #1: Well written article with good methodology and good data analysis. It discusses the prevalence and scope of the oro-maxillary facial surgery in Sudan, which is a developing country. This article also gives us insight as to how we can improve the knowledge about the various oral pathologies and the need to take care of them in a timely fashion.

6. PLOS authors have the option to publish the peer review history of their article (what does this mean?). If published, this will include your full peer review and any attached files.

Reviewer #1: **Yes: **Priyanka Bhandari

---

## [Author Response · Author response to Decision Letter 0]

14 Feb 2021

1. The manuscript meets PLOS ONE's style requirement.

2.The data of the study was shared as supporting infromation excel file.

---

## [Editor Report · Decision Letter 1]

12 Mar 2021

Referral pattern of oral and maxillofacial surgery cases in Sudan: A retrospective age- and sex-specific analysis of 3,478 patients over four years

PONE-D-20-35051R1

Dear Dr. Osman,

We’re pleased to inform you that your manuscript has been judged scientifically suitable for publication and will be formally accepted for publication once it meets all outstanding technical requirements.

Kind regards,

Amit Sapra

Academic Editor

PLOS ONE
---

## [Editor Report · Acceptance letter]

19 Mar 2021

PONE-D-20-35051R1 

Referral pattern of oral and maxillofacial surgery cases in Sudan: A retrospective age- and sex-specific analysis of 3,478 patients over four years 

Dear Dr. Osman:

I'm pleased to inform you that your manuscript has been deemed suitable for publication in PLOS ONE. Congratulations! Your manuscript is now with our production department. 

Kind regards, 

on behalf of

Dr. Amit Sapra 

Academic Editor

PLOS ONE